# Meta-analysis of nationwide SARS-CoV-2 infection fatality rates in India

**Lauren Zimmermann**[1,2]*, **Bhramar Mukherjee**[1,2,3]

**1** Department of Biostatistics, University of Michigan, Ann Arbor, Michigan, United States of America,
**2** Center for Precision Health Data Science, University of Michigan, Ann Arbor, Michigan, United States of America, **3** Department of Epidemiology, University of Michigan, Ann Arbor, Michigan, United States of America

* lzimm@umich.edu

## Abstract

There has been much discussion and debate around underreporting of deaths in India in media articles and in the scientific literature. In this brief report, we aim to meta-analyze the available/inferred estimates of infection fatality rates for SARS-CoV-2 in India based on the existent literature. These estimates account for uncaptured deaths and infections. We consider empirical excess death estimates based on all-cause mortality data as well as disease transmission-based estimates that rely on assumptions regarding infection transmission and ascertainment rates in India. Through an initial systematic review (Zimmermann et al., 2021) that followed PRISMA guidelines and comprised a search of databases PubMed, Embase, Global Index Medicus, as well as BioRxiv, MedRxiv, and SSRN for preprints (accessed through iSearch) on July 3, 2021, we further extended the search verification through May 26, 2022. The screening process yielded 15 studies qualitatively analyzed, of which 9 studies with 11 quantitative estimates were included in the meta-analysis. Using a random effects meta-analysis framework, we obtain a pooled estimate of nationwide infection fatality rate (defined as the ratio of estimated deaths over estimated infections) and a corresponding confidence interval. Death underreporting from excess deaths studies varies by a factor of 6.1–13.0 with nationwide cumulative excess deaths ranging from 2.6–6.3 million, whereas the underreporting from disease transmission-based studies varies by a factor of 3.5–7.3 with SARS-CoV-2 related nationwide estimated total deaths ranging from 1.4–3.4 million, through June 2021 with some estimates extending to 31 December 2021. Underreporting of infections was found previously (Zimmermann et al., 2021) to be 24.9 (relying on the latest 4th nationwide serosurvey from 14 June-6 July 2021 prior to launch of the vaccination program). Conservatively, by considering the lower values of these available estimates, we infer that approximately 95% of infections and 71% of deaths were not accounted for in the reported figures in India. Nationwide pooled infection fatality rate estimate for India is 0.51% (95% confidence interval [CI]: 0.45%– 0.58%). We often tend to compare countries across the world in terms of total reported cases and deaths. Although the US has the highest number of *reported* cumulative deaths globally, after accounting for underreporting, India appears to have the highest number of cumulative total deaths (*reported + unreported*). However, the large number of estimated infections in India leads to a lower infection fatality rate estimate than the US, which in part is due to the younger population in India. We

**Data Availability Statement:** Data and code files are available as Supporting Information files in S1 Data and S1 Code, respectively.

**Funding:** LZ and BM were supported by funding from the University of Michigan School of Public Health and Center for Precision Health Data Science. The funders had no role in study design, data collection and analysis, decision to publish, or preparation of the manuscript.

**Competing interests:** The authors have no competing interests.

emphasize that the age-structure of different countries must be taken into consideration while making such comparisons. More granular data are needed to examine heterogeneities across various demographic groups to identify at-risk and underserved populations with high COVID mortality; the hope is that such disaggregated mortality data will soon be made available for India.

## Introduction

The second wave of SARS-CoV-2 in the 2$^{nd}$ most populous country in the world, India, registered 414 thousand daily cases and 4.5 thousand daily deaths at its peak in May of 2021 [2], and led to a collapse of healthcare infrastructure [3]. Multiple studies indicate that the true number of infections and deaths are orders of magnitude larger [1, 4, 5]. Considerable effort has been devoted towards investigating the true number of SARS-CoV-2 attributed deaths and inferred infection fatality rates (IFR) in India. This brief report systematically synthesizes the existent literature on the true SARS-CoV-2 IFR in India (as of 26 May 2022), through a meta-analysis of studies based on excess deaths and studies based on epidemiological disease transmission models that present relevant estimates through at least June 2021, capturing most of the second wave in India.

## Methods

In brief, we describe the systematic review framework that has previously been detailed in full with the complete search strategy [1]. Adhering to PRISMA guidelines (Table A in S1 Text includes the PRISMA checklist), the databases PubMed, Embase, Global Index Medicus, as well as BioRxiv, MedRxiv, and SSRN for preprints (accessed through iSearch), were searched on July 3, 2021 and results were updated through May 26, 2022. Using this approach, 4,971 citations were screened resulting in **15** studies classified into the following three groups: excess deaths studies (9 articles), disease transmission-based studies estimating unreported deaths (5 articles), disease transmission-based studies using reported deaths only (1 article). Since the three groups are not directly comparable, among the 15 studies, the **9** excess deaths studies with **11** datapoints are included in the nationwide quantitative synthesis. We were unable to stratify and separately meta-analyze disease transmission-based estimates (less than 3 studies rendered through at least June 2021 in the search verification). Several measures of fatality have been used in the literature as indicated in the glossary box. Using a random effects model with DerSimonian-Laird estimates and corresponding confidence intervals (CI), we meta-analyze IFR$_2$ (defined as the infection fatality rate that accounts for death underreporting, as well as case underreporting). We provide a pooled estimate of nationwide IFR$_2$ for SARS-CoV-2 in India with corresponding 95% CI. While this meta-analysis focuses on nationwide studies in India, we summarize the **18** other subnational/regional studies (not meta-analyzed) in Table B in S1 Text. A detailed explanation of the meta-analysis framework is provided in Methods B in S1 Text and Methods C in S1 Text, including Fig A in S1 Text displaying the process from data extraction to obtaining meta-analyzable IFRs.

Lastly, ethical approval is not applicable to the present study. The research uses publicly available data, and is IRB exempt.

## Results

Fig 1 displays the PRISMA flow diagram reflecting the number of included articles from the updated search verification through May 26, 2022. For India countrywide, underreporting

Glossary

$$CFR = \frac{Reported\ Cumulative\ Deaths (at\ a\ 14\ day\ lag)}{Reported\ Cumulative\ Cases}$$

$$Excess\ Deaths = Observed\ All\ Cause\ Mortality - Expected\ All\ Cause\ Mortality$$

$$URF\ (C) = \frac{\textbf{Estimated}\ Total\ Cumulative\ Infections}{Reported\ Cumulative\ Cases}$$

$$URF\ (D) = \frac{\textbf{Estimated}\ Total\ Cumulative\ Deaths}{Reported\ Cumulative\ Deaths\ (at\ a\ 14\ day\ lag)}$$

$$IFR_1 = \frac{Reported\ Cumulative\ Deaths\ (at\ a\ 14\ day\ lag)}{\textbf{Estimated}\ Total\ Cumulative\ Infections}$$

$$IFR_2 = \frac{\textbf{Estimated}\ Total\ Cumulative\ Deaths}{\textbf{Estimated}\ Total\ Cumulative\ Infections}$$

factors (URF) for deaths based on excess deaths studies range from 6.1–13.0 with cumulative excess deaths ranging from 2.6–6.3 million (as shown in Table 1). Considering estimates from disease transmission-based studies, URF ranges from 3.5–7.3 for India with total estimated deaths attributed to SARS-CoV-2 ranging from 1.4–3.4 million (see Table 1). As previously reported [1], URF for cases/infections (inferred from the most recent seroprevalence estimate) is 24.9 using the $4^{th}$ nationwide serosurvey [6]. As such, the evidence suggests that even by the lowest of these estimates roughly 95% of cases (URF (Case) is reportedly 24.9) and 71% of deaths (URF (Death) is at least 3.5) were missed in India.

Nationwide pooled $IFR_2$ estimate for India is 0.51% (95% confidence interval [CI]: 0.45%–0.58%), as presented in Fig 2. This estimate attributes 100% of excess deaths to SARS-CoV-2 during 2020–2021. In actuality, the proportion of excess deaths resulting from COVID-19 is not likely to wholly account for the total excess deaths during the pandemic period, and as such this estimate of 0.51% is likely an overestimate. However, disease transmission-based studies give us a nationwide pooled $IFR_2$ estimate of 0.34% (95% CI: 0.28%– 0.41%), although we caution that this second estimate relies on less than 3 data points. Overall, comparing $IFR_2$ to the nationwide pooled $IFR_1$ (calculated based on reported deaths) of 0.10% (95% CI: 0.07%– 0.14%) [1], we find that $IFR_2$ is roughly 4 times greater than $IFR_1$. Lastly, Fig B in S1 Text presents a visualization of the publication bias assessment among the included studies, and the Egger and Begg tests for asymmetry, as well as the Joanna Briggs Institute (JBI) risk of bias results are presented in the supplementary content (see Methods D in S1 Text and Methods E in S1 Text).

## Discussion and conclusions

Over two years since the start of the pandemic, numerous peer-reviewed studies have focused on understanding the actual death toll of SARS-CoV-2 in India, primarily either via excess

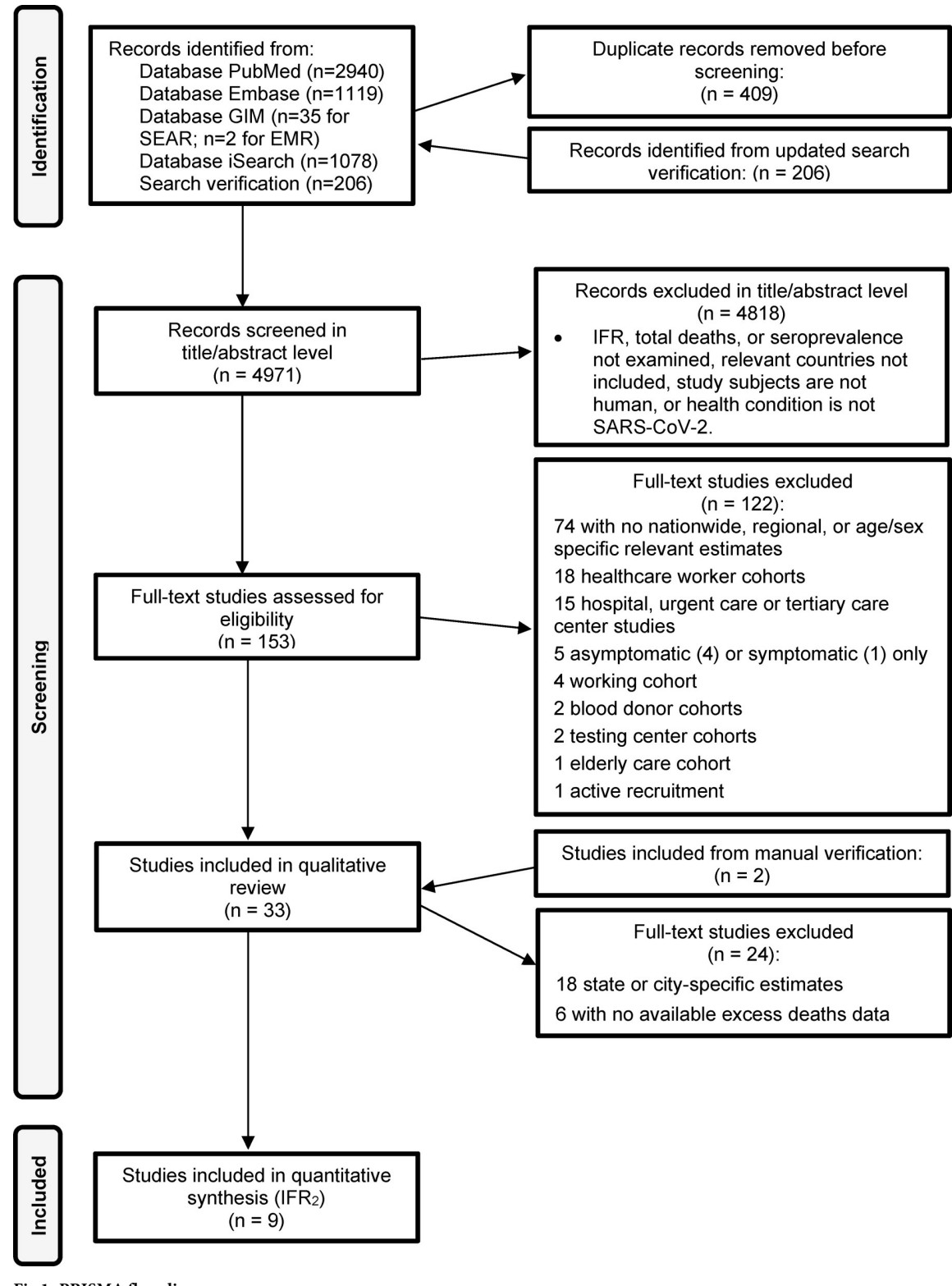

**Fig 1. PRISMA flow diagram.**

**Table 1. Summary of nationwide mortality data from included studies in India from 2020–2021.** Seroprevalence of 67.6% is used with 765 million infections[a] from an age-adjusted population as of 14 Jun-6 Jul 2021 from the 4th nationwide serosurvey [6].

| Study | | Time Period | Estimated Total Deaths (LL, UL) in Millions | COVID-19 Reported Deaths[1] | Under Reporting Factor (LL, UL)[2] | Data Source(s) | Infection Fatality Rate (%) |
|---|---|---|---|---|---|---|---|
| *Excess Deaths Studies* | | | | | | | |
| Wang et al., 2022[3] [5] | * | Jan '20-Dec '21 | 4.0 (95% UI: 3.7, 4.3) | 481,080 | 8.3 (7.5, 8.9) | CRS | 0.52 |
| World Health Organization & Knutson, 2022 [9] | * | Jan '20-Dec '21 | 4.7 (95% CI: 3.3, 6.4) | 481,080 | 9.8 (6.8, 13.4) | Human Mortality Database, World Mortality Dataset, ACM subnational data | 0.61 |
| The Economist & Solstad, 2021 [12] | * | Jan '20-Dec '21 | 4.8 (95% CI: 1.2, 8.2) | 481,080 | 10.1 (2.6, 17.2) | Human Mortality Database, World Mortality Dataset, ACM subnational data | 0.63 |
| Jha et al., 2022[3] [4] | | July '20-May '21 | 630 th (95% CI: 531, 730) | N/A | N/A | Facility-based deaths sample from HMIS | – |
| | | July '20-May '21[5] | 1.2 (95% CI: 1.0, 1.4) | 204,330 | 6–7 | CRS | 0.15 |
| | * | Jun '20-Jul '21[6] | 3.2 (95% CI: 3.1, 3.4) | 450,000 | 6–7 | CVoter | 0.42 |
| Anand et al., 2021[4] [14] | * | Apr '20-Jun '21 | 3.4 (range:1.1, 4.0) | 400,000 | 8.5 (2.7, 10.0) | CRS | 0.44 |
| | * | Apr '20-Jun '21 | 4.0 | 400,000 | 10.0 | International age-specific infection fatality rates | 0.52 |
| | * | Apr '20-Jun '21 | 4.9 | 400,000 | 12.2 | CMIE | 0.64 |
| Guilmoto, 2022[3] [15] | | Mar '20-May '21 | 3.2 | 458,900 | 7.0 | Indian Railways, Kerala age & sex-specific death rates | 0.41 |
| | * | Mar '20-Nov '21 | 3.7 | 458,900 | 8.6 | MLA, Kerala age & sex-specific death rates | 0.48 |
| Leffler et al., 2022[4] [16] | * | Jan '20-Aug '21 | 2.6 (range:1.9, 3.5) | 438,560 | 6.1 (4.5, 8.1) | CRS | 0.34 |
| Malani & Ramachandran, 2021[3] [17] | * | Feb '20-Aug '21 | 6.3 | 458,470 | 13 | CMIE | 0.82 |
| Banaji & Gupta, 2021[3] [18] | * | Apr '20-Jun '21 | 3.8 (range:2.8, 5.2) | 399,489 | 9.5 (6.9, 13.0) | CRS | 0.50 |
| **Study** | | **Time Period** | **Estimated Total Deaths (LL, UL) in Millions** | **COVID-19 Reported Deaths[1]** | **Under Reporting Factor (LL, UL)[2]** | **Data Source(s)** | **Infection Fatality Rate (%)** |
| *Disease Transmission-based Studies* | | | | | | | |
| *Using Reported and Unreported COVID-19 Deaths* | | | | | | | |
| Barber et al., 2022 [7] | | Jan '20-Nov '21 | 3.4 (95% UI: 2.5, 4.9) | 470,810 | 7.3 (5.3, 10.4) | COVID-19 reported cases and deaths from covid19india.org, nationwide and state level serosurveys, hospitalizations from IDSP, excess death estimates from Wang et al. (2022) | 0.3 (95% UI: 0.3, 0.5) |

(*Continued*)

**Table 1.** (Continued)

| Study | Time Period | Estimated Total Deaths (LL, UL) in Millions | COVID-19 Reported Deaths[1] | Under Reporting Factor (LL, UL)[2] | Data Source(s) | Infection Fatality Rate (%) |
|---|---|---|---|---|---|---|
| Zimmermann et al., 2021 [1] | Apr '20-Jun '21 | 1.4 (95% CrI: 1.3, 1.4) | 412,019 | 3.5 | COVID-19 reported cases and deaths from covid19india.org, COVID-19 infections from nationwide serosurvey | 0.36 (95% CrI: 0.35, 0.38) |
| Rahmandad et al., 2021[7] [19] | Jan-Dec 2020 | N/A | N/A | N/A | COVID-19 reported cases and deaths from JHU CSSE, testing data from Indian Council of Medical Research, World Bank indicators | 0.35 (95% CrI: 0.32, 0.39) |
| Shewade et al., 2021 [20] | Jan-Jul 2020 | 197 th | 173,153 | 5.5–11.0 | COVID-19 reported cases and deaths from worldometers.info/coronavirus, CRS deaths registration coverage and errors in MCCD | 0.58–1.16 |
| Campbell & Gustafson, 2021 [21] | May-Jun 2020 | 46 th | 12,573 | 3.6 | COVID-19 reported deaths from ourworldindata.org/coronavirus/country/india, COVID-19 infections from nationwide serosurvey, death underreporting factor from Purkayastha et al. (2021), WDI nationwide age proportions | 0.29 (95% CrI: 0.09, 0.90) |
| *Using Reported COVID-19 Deaths* | | | | | | |
| Song et al., 2021 [22] | Mar '20-Jun '21 | 532 th (95% CI: 513, 552) | 399,489 | 1.3 (1.2, 1.4) | COVID-19 reported cases and deaths from WHO COVID-19 Dashboard, Influenza reported cases from WHO FLUNET | 0.06 |

*Notes*: Asterisk (\*) denotes that the excess deaths study was included in the quantitative meta-analysis, being through at least June 2021.

N/A = Not available, CRS = Civil Registration System, MLA = Member of the Legislative Assembly sample, CVoter = CVoter India Omnibus telephone survey, HMIS = Health Management Information System, ACM = all-cause mortality, CMIE = Center for Monitoring Indian Economy Consumer Pyramids Household survey, IDSP = Integrated Disease Surveillance Programme (for Goa, India). JHU CSSE = Johns Hopkins University Center for Systems Science and Engineering, MCCD = Medical Certification of Cause of Death from Ministry of Home Affairs, WDI = World Bank's World Development Indicators. Lower and upper uncertainty bounds for all-cause excess deaths estimates are included in this table, when provided in the study.

[a] Estimated total cumulative infections is calculated as the seroprevalence of 67.6% among ages ≥ 6 years from the latest 4th nationwide serosurvey study in India [6] multiplied by the age-adjusted population (additional details are included in Methods B in S1 Text and Fig A in S1 Text).

[1] COVID-19 Reported Deaths are obtained from covid19india.org, unless otherwise noted.

[2] Underreporting Factor is computed as Excess Deaths divided by COVID-19 Reported Deaths, unless otherwise noted.

[3] Underreporting Factor (URF), as well as COVID-19 Reported Deaths are directly reported in this study. Hence, the URF in this table is the precalculated estimate provided.

[4] Excess Deaths, as well as COVID-19 Reported Deaths, are directly reported in this study.

[5] The COVID-19 Reported Deaths provided in this study are across select states in the Civil Registration System (CRS).

[6] The precalculated Underreporting Factor and COVID-19 Reported Deaths reported in this study are through September 2021.

[7] Numerical estimates for total deaths are unavailable for Rahmandad et al. (2021) [19], and are thereby displayed as not available in this table.

deaths or disease transmission-focused modeling, enabling the meta-analysis herein of the 11 identified excess deaths estimates. When appropriately accounting for case and death underreporting, the cumulative SARS-CoV-2 infection fatality rate in India varies within a 95% CI of 0.45%-0.58%, which indicates that $IFR_2$ is 4–6 times more than what is being reported based on tabulated deaths due to COVID-19. The disease transmission-based estimates qualitatively appear to be more conservative than the ones that originated from excess deaths studies. One possible explanation could stem from the fact that most of the excess death studies are based on all-cause-mortality data and do not quantify the proportion of the excess deaths attributable to COVID-19. The pooled $IFR_2$ estimate from COVID-specific transmission model-based studies is largely congruent to the estimated IFR of 0.3% (as of 14 November 2021) for India reported in the global IFR study by Barber et al. (2022) [7].

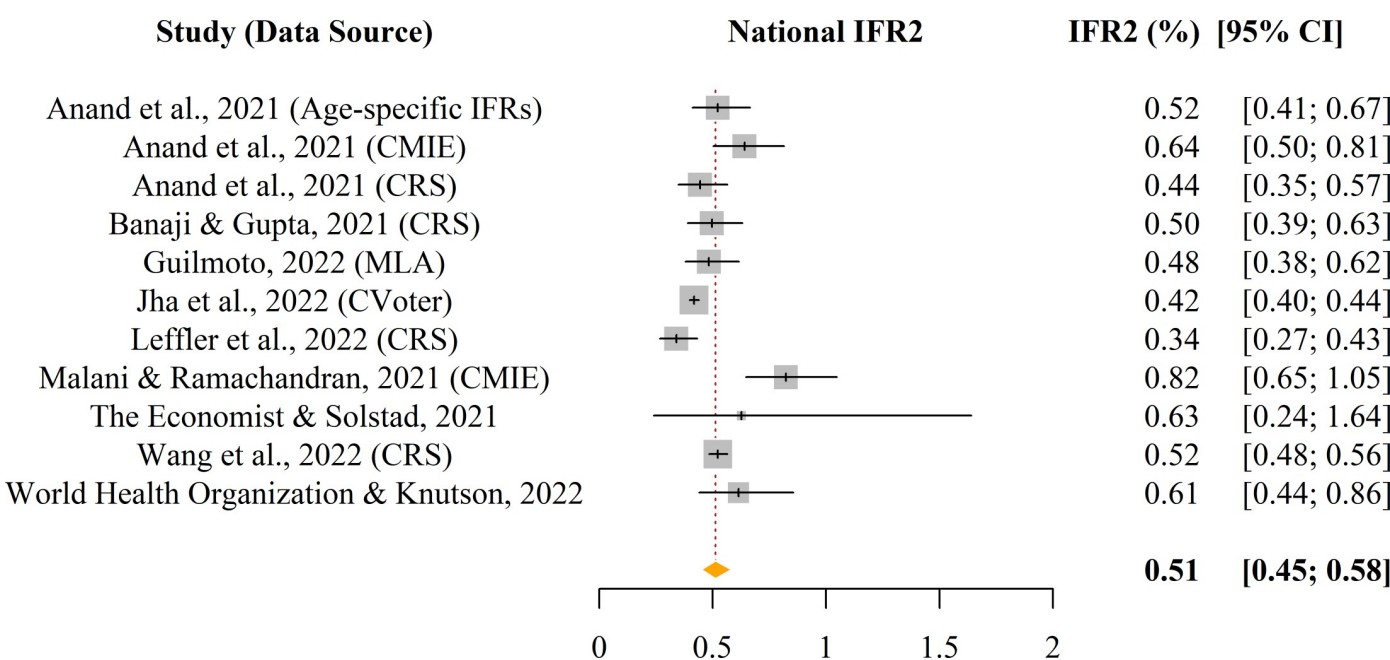

**Fig 2. Nationwide estimated pooled IFR$_2$ of SARS-CoV-2 for India, through June 2021 and extending to December 2021.** Included studies listed in this forest plot are categorized as excess deaths studies. There were too few disease transmission-based studies (less than 3 studies through June 2021) identified by the search, and therefore, further meta-analyzing the additional category of disease transmission-based studies was not feasible.

Limitations of the meta-analysis are as follows. First, insufficient data on age and sex-disaggregated mortality for India precluded investigation into heterogeneity by such demographics. Second, multiple studies rely on excess deaths derived from common sources, such as the civil registration system (CRS) data, as well as infections derived from nationwide serosurveys, which rules out independence between included studies and may bias the resulting pooled estimate. India recently released the CRS data for 2020, but most studies estimate largest excess deaths during April-June of 2021 and no CRS data are available for this period. Moreover, the incompleteness of CRS data may hinder representativeness and, thereby, complicates the interpretability of excess deaths estimates relying on CRS data. The more nationally representative sample registration system (SRS) is often used to adjust for missing death information in CRS, but SRS data are not yet available for 2020 and 2021. Lastly, while we use the latest available nationwide serosurvey to obtain an age-adjusted infections estimate in computing the IFR for SARS-CoV-2, we acknowledge that this approach does not incorporate factors of waning immunity and re-infections. If such components were able to be accounted for, the denominator of the IFR (estimated infections) may have been larger and thereby the true IFR will be attenuated to a degree. Such limitations inherent to sero-surveillance studies also include sero-reversion which concerns reduced detection of SARS-CoV-2 antibodies and leads to an upward bias in IFR estimates [8].

It is critical to contextualize the uncaptured SARS-CoV-2 infections and deaths in India, and how such underreporting could distort comparisons of disease spread and mortality within countries across the world. Considering the three countries with the highest cumulative reported deaths (as of December 31, 2021), namely, India, Brazil, and the United States (in ascending order), the IFR$_2$ (as of 14 November 2021) reported by Barber et al. (2022) appears to be the lowest in India (IFR$_2$ of 0.3%) compared to the US (IFR$_2$ of 0.9%) and Brazil (IFR$_2$ of 0.5%) [7]. This is due to the very large number of estimated cumulative infections in India (approximately 1 billion, through mid-November 2021 [7]). With respect to the total number of deaths, Wang et al. (2022) estimate deaths to be underreported by a factor of 8.3, 1.3, and

1.2 for India, the US, and Brazil, respectively [5]. This is qualitatively similar to the death underreporting factors reliant on WHO estimates (similarly through 31st December 2021) of 9.8, 1.1, and 1.1 for India (4.7 million excess deaths and 481,080 reported deaths), the US (933,547 excess deaths and 818,464 reported deaths), and Brazil (681,514 excess deaths and 618,817 reported deaths), respectively [9]. These rankings indicate that underreporting of deaths (through 31st December 2021) is particularly acute for India.

While metrics are useful for evaluating public health policies, we caution against such crude comparisons based on a single metric. Although we use cumulative excess deaths as a measure of comparison in mortality ranking, population counts are not factored in and deaths per million may be preferable in another context. In addition, such overall mortality comparisons must be placed in the context of the age-structure of the different countries. India has a younger population (Median age 28 years) than the US (Median age 38 years) or Brazil (Median age 34 years) [10]. Age-specific $IFR_2$ should be used, if possible, when examining COVID-19 mortality burden within and across countries and in subsequent decision making. Recent studies underscore the importance of adjusting for age structures, when performing related deaths estimations. For example, *The Economist* recently made available an age-adjusted IFR source [11], which is further incorporated into their published estimates [12]. Disaggregated mortality data are necessary to validate these age-specific estimates for India.

Many of the included studies in this meta-analysis also sought to account for changes in mortality and subsequently changes in IFR over time often by incorporating as granular, longitudinal data as possible. This is important as the lethality of the virus is subject to multiple time-varying components, especially the roll-out of vaccines (starting in January 2021 within India), as well as the changing variant landscape wherein the milder SARS-CoV-2 variant Omicron and sub-lineages became dominant.

We look forward to the release of timely, disaggregated data on SARS-CoV-2 deaths within India to assess the burden of COVID-19 among various demographic groups [13], as well as to enable targeted policy interventions. Once nationwide 2021 CRS reports are released, the findings with respect to the excess death estimates will be further validated. In the absence of data, we must rely on curated estimates computed by multiple teams of dispassionate scientists and a systematic review and synthesis of such evidence.

## Supporting information

**S1 Table. Results of risk of bias assessment for included articles.**
(XLSX)

**S1 Text.**
(DOCX)

**S1 Data.**
(XLSX)

**S2 Data.**
(R)

**S1 Code.**
(R)

## Acknowledgments

The authors thank the information specialists from the University of Michigan Taubman Health Sciences Library for their prior guidance on the search strategy for the systematic

review that enabled this meta-analysis. The authors also wish to thank Maxwell Salvatore for providing technical advice and feedback regarding the graphics in this report.

## Author Contributions

**Conceptualization:** Lauren Zimmermann, Bhramar Mukherjee.

**Data curation:** Lauren Zimmermann.

**Formal analysis:** Lauren Zimmermann, Bhramar Mukherjee.

**Investigation:** Lauren Zimmermann, Bhramar Mukherjee.

**Methodology:** Lauren Zimmermann, Bhramar Mukherjee.

**Supervision:** Bhramar Mukherjee.

**Writing – original draft:** Lauren Zimmermann, Bhramar Mukherjee.

**Writing – review & editing:** Lauren Zimmermann, Bhramar Mukherjee.

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
