## [Decision Letter · Decision Letter 0]

26 Apr 2022

PGPH-D-21-01072

Meta-analysis of nationwide SARS-CoV-2 infection fatality rates in India

Dear Lauren Zimmermann,

Thank you for submitting your manuscript to PLOS Global Public Health. After careful consideration, we feel that it has merit but does not fully meet PLOS Global Public Health’s publication criteria as it currently stands. Therefore, we invite you to submit a revised version of the manuscript that addresses the points raised during the review process.

Please consider all issues mentioned in the reviewers' comments. While they seem to agree this is an important work, the reviewrs do raise some significant issues in the methodology of the manuscript. Please indicate every change made in response to their comments and provide justifications for any comments not addressed. 

Please submit your revised manuscript by . If you will need more time than this to complete your revisions, please reply to this message or contact the journal office at globalpubhealth@plos.org. Please include the following items when submitting your revised manuscript:

We look forward to receiving your revised manuscript.

Kind regards,

Jong-Hoon Kim, Ph.D.

Academic Editor

Journal Requirements:

2. Please provide separate figure files in .tif or .eps format and remove the embedded figures from the manuscript file.

Additional Editor Comments (if provided):

Reviewers' comments:

Reviewer's Responses to Questions

**Comments to the Author**

1. Does this manuscript meet PLOS Global Public Health’s publication criteria? Is the manuscript technically sound, and do the data support the conclusions? The manuscript must describe methodologically and ethically rigorous research with conclusions that are appropriately drawn based on the data presented.

Reviewer #1: Partly

Reviewer #2: Yes

Reviewer #3: Yes

Reviewer #4: Yes

2. Has the statistical analysis been performed appropriately and rigorously?

Reviewer #1: I don't know

Reviewer #2: Yes

Reviewer #3: I don't know

Reviewer #4: Yes

3. Have the authors made all data underlying the findings in their manuscript fully available (please refer to the Data Availability Statement at the start of the manuscript PDF file)?

Reviewer #1: Yes

Reviewer #2: Yes

Reviewer #3: Yes

Reviewer #4: Yes

4. Is the manuscript presented in an intelligible fashion and written in standard English?

Reviewer #1: Yes

Reviewer #2: Yes

Reviewer #3: Yes

Reviewer #4: Yes

5. Review Comments to the Author

Reviewer #1: The paper is potentially a valuable addition to the literature on the impact of the pandemic in LMICs. In a context of hugely variable surveillance and major gaps in our knowledge of the impact of the pandemic in the Global South, such work is of high importance. Given India's sheer size and weak surveillance, understanding pandemic mortality in India is perhaps the most critical challenge to understanding pandemic mortality worldwide.

However, there are some limitations to the work, which the authors would do well to address. First some general observations and questions:

It would be helpful if the 15 studies which were included in the meta-analysis are listed clearly, perhaps in a table, along with an outline of what kind of data or analysis appears in each, and how this data is used to arrive at IFR estimates.

The eligibility criteria for included and excluded papers (Figure 2) are not clear - it appears that papers which either model pandemic mortality or include prevalence or mortality data were included. Were papers focussing on subnational data excluded, or were no such papers found to meet the eligibility criteria? In sum, the framework needs to be spelled out more clearly - Figure 2 does not seem adequate.

In reading a meta-analysis there is often the assumption in the reader's mind - rightly or wrongly - that the studies are in some sense "independent". They may involve different data-sets, different geographical regions, different methodologies, etc. Here, however, it appears that many of the estimates are based around overlapping data and techniques. This applies for example to the estimates of excess mortality based on civil registration data, and also perhaps to estimates of prevalence based on serosurveys. Overall, it appears that there is relatively little "independent" data behind the numbers from different papers, increasing the risk of systematic bias (more on this below). This warrants at least some discussion.

The study includes both data- and model-based estimates, and it is observed that the model-based estimates of IFR are more conservative than data-based ones. However, model-based estimates reflect a complex array of assumptions - including, perhaps, about ascertainment of infections and deaths, and even sometimes about IFR itself. This leads to the possibility of circularity. Without some discussion of the assumptions behind the models it is very hard to interpret model-based IFR estimates, or to know whether they deserve the same weight as data-based ones. (Although data-based estimates also rely on a host of assumptions, these are in general more obvious to see). This seems to me to be of critical importance in a meta-analysis which relies on a small number of papers, a sizeable fraction of which are modelling papers.

On the theme of model-based estimates, disproportionate weight appears to be given in the text to the IHME modelling, whose estimates appear to be outliers even amongst the model-based estimates (and certainly amongst the estimates from all sources). Are the IHME projections available in any kind of preprint, perhaps submitted for peer-review? Is this work sufficiently transparent to meet the criteria for this kind of analysis? (I have some doubts about this, but these would be allayed if the authors referenced a technical document which included details of how the IHME estimates are arrived at, specifically for India.)

Overall, potential systematic biases deserve much more discussion. For example, in estimates based on survey or civil registration data: the possibility of non-COVID excess mortality, particularly during periods when healthcare infrastructure is overwhelmed; potential sampling biases in survey-based estimates; issues around incompleteness of death registration and possible differential impact of the pandemic on communities where registration is higher/lower; possible issues with the fact that available data is often from sources (e.g. online systems) which are incomplete. Although a meta-analysis may not be the place to discuss in detail potential biases in each study, it is important for readers to understand the limitations to the work, which are often common to many papers.

Some specific observations and questions:

What is the relationship between this paper and reference (1) Zimmermann et al? It would be helpful if the authors could clarify whether this paper is a presentation of the key findings of their earlier work, whether it includes further data and analysis, and so forth.

The line: "Supplementary Figure 2 shows a nationwide pooled IFR 2 estimate for India of 0.499%" does not seem to accord with Figure 2, where the pooled estimate appears to be 0.44%.

IFR1 (i.e., IFR based on recorded COVID-19 deaths) is fairly easy to calculate, but the origin of the IFR1 estimates quoted in the paper are not clear. If they are national estimates, then they are too high. For example, taking national level reported COVID-19 deaths, IFR1=0.1% would be consistent with an infection rate of a mere 34%, considerably lower than the estimates from recent serosurveys. The source of this mismatch requires clarification. Data from a few well-surveilled urban localities generally gives much higher values of IFR1 - is this the source of the high IFR1 estimate? If so, the aggregate IFR1 estimate should not be used to provide national estimates of URF (D). These matters are vitally important from the point of view of understanding the scale, geographical variation, and reasons behind COVID-19 death under-reporting in India.

Although the aggregate estimates of IFR2 are plausible, the calculations which underlie the individual estimates are not transparent. For example, some estimates are based on prevalence estimates in Murhekar et al (June 2020); but it is unclear what mortality data has been used as the numerator in these estimates. Do many of the papers include mortality data up to June-July 2020? If so, how reliable are these figures given, for example, the various delays and disruption to death registration during national lockdown? It would seem that a multiplicity of possible errors make the early estimates highly unreliable.

In Table 1, the Wave 2 estimate for Banaji and Gupta is empty, but the preprint does include a point estimate (11.3) of the death underreporting factor during March-May 2021 in the subsection "Increasing under-ascertainment of COVID-19 deaths?" Is there a reason this was omitted?

In Figure 3, an IFR2 estimate for India of 0.135% is given based on IHME projections. But the IHME seemed to estimate 2.8M total deaths on Sept. 20, 2021, and a 66% infection rate, which would together give an IFR2 estimate of around 0.3%. Could the authors clarify the source of the IFR2 estimate of 0.135% attributed to the IHME?

Reviewer #2: The studies reviewed could be updated and expanded:

The following study has India data which could be incorporated:

Wang H, Paulson KR, Pease SA, Watson S, Comfort H, Zheng P, Aravkin AY, Bisignano C, Barber RM, Alam T, Fuller JE. Estimating excess mortality due to the COVID-19 pandemic: a systematic analysis of COVID-19-related mortality, 2020–21. The Lancet. 2022 Mar 10.

Your fourth reference by Deshmukh doesn’t list the title of the report:

Y. Deshmukh, W. Suraweera, C. Tumbe, A. Bhowmick, S. Sharma, P. Novosad, S.

H. Fu, L. Newcombe, H. Gelband, P. Brown, P. Jha, medRxiv, in press,

doi:10.1101/2021.07.20.21260872.

I think that study is the same is this one by Jha, which is now published:

Jha P, Deshmukh Y, Tumbe C, Suraweera W, Bhowmick A, Sharma S, Novosad P, Fu SH, Newcombe L, Gelband H, Brown P. COVID mortality in India: National survey data and health facility deaths. Science. 2022 Jan 6:eabm5154.

The study by Leffler et al is now published in a journal:

Preliminary Analysis of Excess Mortality in India During the COVID-19 Pandemic.

Leffler CT, Lykins V JD, Das S, Yang E, Konda S.

Am J Trop Med Hyg. 2022 Apr 4:tpmd210864. doi: 10.4269/ajtmh.21-0864. Online ahead of print. PMID: 35378508.

The published version of the study by Leffler now has mortality data through August 31, 2021.

The study by Guilmoto is now published:

Christophe Z Guilmoto. An alternative estimation of the death toll of the Covid-19 pandemic in India., PLoS One. 2022 Feb 16;17(2):e0263187. doi: 10.1371/journal.pone.0263187. eCollection 2022.

Reviewer #3: The study appears an important meta-analysis of model and excess-deaths based studies of the pandemic's death toll in India, and using undercounting-ratios of infections based on seroprevalence studies, what a plausible range of the covid-19 IFR in India might be.

I believe that provided my below recommendations are followed and/or comments addressed, the study should be published and would be a valuable contribution. (This is also my understanding of the selection "minor revision"). I defer to discipline specialists on the fitness of article to journal. I also defer to discipline specialists on the statistical methodology employed, and the soundness and completeness of the replication package.

My comments will be restricted to the data used, and a few of the assumptions underlying the IFR estimations, and presentation. My apologies if I missed something obvious, including elements which address the concerns I raise below.

Comments:

1. The Anand et al study has three estimates, but the study as far as I can tell only includes one of these. I would have liked to have seen an explicit justification of the choice of which to use. While I see a clear reason to exclude the IFR-based estimate, I wonder why the estimation based on the civil-state registration system was not included.

2. Acknowledging that I am an interested party, I find it puzzling that The Economist's estimates of excess deaths in India are not included, despite their global model being widely cited in other academic research (including that cited here), and having estimates on both the first and second wave.

3. Sero-prevalence based estimates are problematic beyond the early stages of the pandemic, because sero-reversion means that infections become progressively less probable to be detected, and because re-infections are possible. (They are also problematic in many contexts because of vaccines.) This should be acknowledged explicitly, and the implications discussed.

4. I found it hard to follow where infection estimates were from - I would have liked something like what was done for table 1 except for infection estimates. Table 2 was confusing to me - I would try to improve this presentation.

3. The use of the IHME estimate of infections is problematic in figure 3, because, unless I am mistaken, this estimate is based on an estimated IFR and estimated death rates, meaning the implied IFR is "baked in". The figure is also unclear (that is, if the bars are meant to mutually exclusive).

5. For reasons discussed elsewhere, see e.g. Andrew Gelman (2021), I would consider avoiding reliance on IHME estimates regarding the pandemic in general.

6. My recommendation would be to drop this comparison to other the US and Brazil, and rather here summarize other estimates of IFR (of which there are many - see e.g. Brazeau et al 2020 and the papers citing this study).

7. With regards to discussion on age structure, the following might be of interest: https://github.com/TheEconomist/covid-19-age-adjusted-ifr -- these also figure in The Economist's global model.

8. I find it strange that there is no mention of changes in IFR over time. This has been known to respond to health care capacity, treatment quality changes, vaccines, and variants. This must at a minimum be acknowledged.

Recommendations:

1. Add justification of selection of which Anand et al study estimate to use

2. Clearer presentation of sources for infection estimates, and a discussion of their limitations (sero-reversion being an important one)

3. Include The Economist's estimate for India's death toll

4. Drop figure 3, which is both confusing, is based on a different approach than the rest of the study, and presents information which is already widely known

5. Replace this with a discussion, and perhaps a table, of how this compares to other studies' estimates of IFRs.

6. Consider including a mention of age-adjusted implied IFR by country.

8. Add to discussion the limitations of the study as it relates to varying IFRs over time: this should mention vaccines, the sars-cov-2 variants, treatment quality, and how this study is based on data from a certain such context, and how this context has changed.

Best of luck in revisions of this important research

Reviewer #4: Major Comments

I wasn't totally sure of the methodology, and I see that estimates of the IFR can be obtained using lower bounds on under-reporting. But with a sampling-based approach (from the asymptotic distribution of your estimators), couldn't you obtain estimates of IFR that account for uncertainty in the underreporting?

It wasn't clear to me what was model-based and what wasn't. IHME and The Economist excess mortality estimates for India would be model-based, right?

Specific Comments

Abstract: ``Highly conservative'' could be misinterpreted as for the ratio (the IFR) but you are referring to the underreporting of numerator and denominator, which does not mean the ratio is highly conservative. To be clear, I'm not suggesting you report different IFRs, but am commenting on the wording.

P3 ``top three countries", via what metrics?

P3 ``obscur". Typo.

P3 Give 3 decimal places for 0.44\\% figure.

P4 Is there a reason why for URF(C) you have "Total Cumulative Infections" in the numerator and "Cumulative Cases" in the denominator. Is it important to distinguish between ``infections" and ``cases"? And no ``Total" in the denominator.

Table 1. I didn't fine this table very easy to take in. Having a column labeled ``excess deaths" would help.

6. PLOS authors have the option to publish the peer review history of their article (what does this mean?). If published, this will include your full peer review and any attached files.

**Do you want your identity to be public for this peer review?** For information about this choice, including consent withdrawal, please see our Privacy Policy.

Reviewer #1: No

Reviewer #2: No

Reviewer #3: **Yes: **Sondre Ulvund Solstad

Reviewer #4: No

---

## [Decision Letter · Decision Letter 1]

22 Jul 2022

Meta-analysis of nationwide SARS-CoV-2 infection fatality rates in India

PGPH-D-21-01072R1

Dear Ms Zimmermann,

We are pleased to inform you that your manuscript 'Meta-analysis of nationwide SARS-CoV-2 infection fatality rates in India' has been provisionally accepted for publication in PLOS Global Public Health.

Best regards,

Jong-Hoon Kim, Ph.D.

Academic Editor

There are a couple of minor editorial comments by Reviewer #3. Please address those when you make formatting changes for the final version.

Reviewer Comments (if any, and for reference):

Reviewer's Responses to Questions

**Comments to the Author**

1. If the authors have adequately addressed your comments raised in a previous round of review and you feel that this manuscript is now acceptable for publication, you may indicate that here to bypass the “Comments to the Author” section, enter your conflict of interest statement in the “Confidential to Editor” section, and submit your "Accept" recommendation.

Reviewer #1: All comments have been addressed

Reviewer #3: All comments have been addressed

Reviewer #4: All comments have been addressed

2. Does this manuscript meet PLOS Global Public Health’s publication criteria? Is the manuscript technically sound, and do the data support the conclusions? The manuscript must describe methodologically and ethically rigorous research with conclusions that are appropriately drawn based on the data presented.

Reviewer #1: Yes

Reviewer #3: Yes

Reviewer #4: Yes

3. Has the statistical analysis been performed appropriately and rigorously?

Reviewer #1: Yes

Reviewer #3: I don't know

Reviewer #4: Yes

4. Have the authors made all data underlying the findings in their manuscript fully available (please refer to the Data Availability Statement at the start of the manuscript PDF file)?

Reviewer #1: Yes

Reviewer #3: Yes

Reviewer #4: Yes

5. Is the manuscript presented in an intelligible fashion and written in standard English?

Reviewer #1: Yes

Reviewer #3: Yes

Reviewer #4: Yes

6. Review Comments to the Author

Reviewer #1: The authors have carefully engaged with all the comments and criticisms, and the paper is much improved. The selection criteria for the studies, and methodological approach are presented more transparently, and it is now easier to understand the detail of the manuscript. Within the framework of incomplete data and multiple, overlapping uncertainties, the authors should be congratulated for drawing together, to the extent possible, a "consensus" on the questions of pandemic mortality/IFR in the Indian context. I recommend that the piece be published.

Reviewer #3: Thank for addressing my concerns. Just two tiny quibbles. First, in table 1, it says The Economist's estimate uses Mumbai data. But the full list of subnational data sources is Tamil Nadu State, Madhya Pradesh State, Andhra Pradesh State, Mumbai City, Kolkata City, and (in Indonesia) Jakarta Province. I would encourage "ACM subnational data" instead, or list all of them. Second, "" ext-link-type="uri" xlink:type="simple">github.com/TheEconomist/covid-19-age-adjusted-ifr" could easier be a standard reference to the article in which it was published: https://www.economist.com/graphic-detail/2020/11/16/why-rich-countries-are-so-vulnerable-to-covid-19

Thanks!

Reviewer #4: My comments have been adequately addressed.

7. PLOS authors have the option to publish the peer review history of their article (what does this mean?). If published, this will include your full peer review and any attached files.

**Do you want your identity to be public for this peer review?** For information about this choice, including consent withdrawal, please see our Privacy Policy.

Reviewer #1: No

Reviewer #3: No

Reviewer #4: No
